# The Uterocervical Angle Combined with Bishop Score as a Predictor for Successful Induction of Labor in Term Vaginal Delivery

**DOI:** 10.3390/jcm10092033

**Published:** 2021-05-10

**Authors:** Seung-Woo Yang, Seo-Yeon Kim, Han-Sung Hwang, Hee-Sun Kim, In-Sook Sohn, Han-Sung Kwon

**Affiliations:** 1Department of Obstetrics and Gynecology, Sanggye Paik Hospital, School of Medicine, Inje University, Seoul 01757, Korea; s5635@paik.ac.kr; 2Department of Obstetrics and Gynecology, Kangbuk Samsung Hospital, Sungkyunkwan University School of Medicine, Seoul 03181, Korea; yeonzza@naver.com; 3Division of Maternal and Fetal Medicine, Department of Obstetrics and Gynecology, Research Institute of Medical Science, Konkuk University School of Medicine, Seoul 05030, Korea; hwanghs@kuh.ac.kr (H.-S.H.); 19960011@kuh.ac.kr (I.-S.S.); 4Department of Obstetrics and Gynecology, Dongguk University Ilsan Hospital, Dongguk University, Goyang 10326, Korea; smallkong7@gmail.com

**Keywords:** induction of labor, uterocervical angle, bishop score, cervical length

## Abstract

The objective of this study was to determine the value of uterocervical angle (UCA) in predicting successful induction of labor (IOL) in singleton pregnant women compared to the Bishop score and cervical length (CL). A total of 205 normal term, singleton labor-induction cases were analyzed. Successful IOL was defined as the onset of active labor of induction. A comparative analysis was performed to evaluate the effectiveness of UCA, Bishop score, and CL in predicting IOL. Compared to the non-successful IOL group, the women in the successful IOL group had significantly wider UCA (*p* = 0.012) and higher Bishop score (*p* = 0.001); however, the CL was not significantly different (*p* = 0.130). UCA alone did not perform better than the Bishop score when predicting successful IOL. However, UCA combined with the Bishop score showed higher performance in predicting IOL (combined UCA > 108.4° and favorable Bishop score as sensitivity of 44.6%, specificity of 96.0%, PPV of 96.2%, and NPV of 43.6; combined UCA > 108.4° or favorable Bishop score as sensitivity of 85.7%, specificity of 50.0%, PPV of 78.7%, and NPV of 61.9). In conclusion, UCA combined with Bishop score may be an effective sonographic method for predicting successful IOL.

## 1. Introduction

Induction of labor (IOL) is a common practice in obstetrics and is followed in many countries with rates ranging from 1.4–35% [1,2]. A recent publication revealed that the rate of IOL was 23.4% in the United States, 22.1% in the United Kingdom, 4.4% in African regions, 11.4% in Latin America, and 12.1% in Asian countries [2]. However, failure of IOL is relatively common, occurring in up to 20% of cases [3,4], and may lead to various maternal and fetal complications [5]. Various studies have defined failed IOL differently, i.e., both as no vaginal delivery and inability to achieve the active phase of labor [6]. During pregnancy, while a favorable cervix is vital for successful delivery, an unfavorable cervix increases the need for cesarean sections (CS) [7]. Therefore, it is crucial to evaluate pre-delivery cervical status. The proper selection of labor induction method, such as ripening or oxytocin, depends on cervical status, and the evaluation of a favorable cervix is a key issue in clinical obstetrics [8,9].

In previous studies, various risk factors have been suggested to predict successful IOL, such as the Bishop score, nulliparity, gestational age, size for gestational age, and maternal obesity [10,11,12]. The Bishop score, calculated subjectively through a pelvic examination, remains widely used for prediction. The Bishop score is determined based on dilatation, effacement, station, cervical consistency, and position [13]. However, several studies commented on its limitation in terms of reproducibility as well as patient discomfort and possible risks of infection or rupture of the membrane [14,15,16]. Given these limitations, better alternatives for IOL prediction are needed.

Some transvaginal sonographic (TVS) parameters appear promising as candidates for predicting IOL. Although TVS requires proper training for measurement, it is preferred over the traditional Bishop score due to its reproducibility [7]. Some parameters have been investigated in previous studies. These include cervical length (CL; measured by transvaginal ultrasonography), cervical stiffness (measured by elastography), angle of progress, and posterior cervical angle [17,18,19,20].

The uterocervical angle (UCA) can also be determined with transvaginal ultrasonography and is defined as the angle between the lower uterine segment and the cervical canal. UCA has recently been explored as an ultrasound parameter that may predict preterm birth [21,22]. The fundamental physics, using the summation of vectors in both directions from the anterior wall and endocervical canal, suggest a relationship between UCA and the prediction of labor [8]. The force exerted by the uterus on the cervix varies depending on the UCA. In a recent study, Dagdeviren et al. reported that patients with broader pre-induction UCAs were prone to have a shorter duration of the active phase [7]. Although a recent study reported that UCA combined with CL predicted labor induction more satisfactorily than CL alone, UCA is more reproducible than CL [21,23,24]. However, as an ultrasound parameter established relatively recently, only a few studies have examined the use of UCA in full-term pregnancies to predict IOL.

Therefore, this study aimed to determine the value of UCA in predicting the risk of induction failure in singleton pregnant women and compare it to CL and the Bishop score.

## 2. Materials and Methods

### 2.1. Patients

This retrospective study included women with a singleton pregnancy between 37 0/7 to 41 0/7 weeks of gestational age at Konkuk University Medical Center between September 2017 and September 2019. A total of 205 singleton pregnancy, pre-induction cervical assessments included the Bishop score, CL, and UCA. Maternal demographic data and delivery information were collected and evaluated, and comparative analysis was performed for the predictive effectiveness of UCA and CL vs. Bishop score. Ethical approval was obtained from the Institutional review board of the Konkuk University Hospital [Ref No: KUH1040063].

### 2.2. Uterocervical Angle Measurement

CL was obtained from the internal os to the external os in a straight line, and the shortest of three measurements was used for analysis. The UCA angle was measured as previously described by Dziadosz et al. [19]. Briefly, the first-angle caliper was placed from the external to the internal orifice of the uterus, and the second caliper was then extended along the length of the lower uterine segment. (Figure 1) For UCA measurement, a sonographer was used to evaluate all cases to ensure standardization of measurement.

### 2.3. Labor Induction

For induction of labor, the standard process was performed. Briefly, IOL began by prostaglanding E2 (dinoprostone) in < 4 of Bishop score and intravenous oxytocin or if >4 of Bishop score or onset of active labor occurred, oxytocin was maintained until delivery [25,26]. The active labor was defined by ≥4 cm of dilatation with regular contraction. Otherwise, intravenous oxytocin was stopped after 12 h, and the procedure was repeated the following day. Successful IOL was defined as the onset of active labor within 12 h. Fetal heart rates were continuously monitored with a central electronic system in all cases.

### 2.4. Statistical Analysis

Data were analyzed using the statistical software package SPSS (version 18.0; SPSS Inc., Chicago, IL, USA). Median values are used to describe continuous data, with discrete variables displayed as totals and frequencies. For univariate analyses, Mann–Whitney U tests were used to compare continuous data. Chi-square tests or Fisher’s exact tests were used for categorical variables as appropriate. A receiver-operating characteristics (ROC) curve was used to determine the optimal cut-off of cervical parameters for the prediction of successful vaginal delivery, and the area under the curve (AUC) was derived. Sample size calculation was done by G*Power 3.0 with 95% of difference, power of 0.8 [27].

## 3. Results

Medical records from a total of 686 cases were reviewed. Inclusion criteria for the study were singleton pregnancy, intact membrane, cephalic presentation, and absence of active labor at admission. A total of 358 women with congenital fetal anomalies, intrauterine fetal death, contraindications to vaginal delivery, previous cervical surgery, refusal to participate, active labor at admission, and fetal distress before active labor were excluded. Additionally, 123 women with the absence of pre-induction transvaginal sonographic data were excluded. (Figure 2) A total of 205 cases of labor induction were included after considering the exclusion criteria that were described above. Clinical characteristics of the study population are presented in Table 1. In indications for IOL, elective induction at ≥39 weeks was the most common indication at 57.6% (*n* = 118). According to the success of IOL criteria, 140 pregnancies were categorized into the success group, and 65 pregnancies were classified as non-successful.

There was no statistically significant difference between the two groups in terms of maternal age, body mass index (BMI), and gestational age at delivery. Cesarean delivery rate was significantly higher in the non-success group. Among the cervical parameters, the Bishop score confirmed on the day of admission showed a statistically significant difference between the success and non-success group (*p* = 0.001). Further, UCA was significantly higher in the success group compared to the non-success group (*p* = 0.012); however, CL was not significantly different (*p* = 0.130); Figure 3.

A ROC curve was used to determine the efficacy of UCA as a predictor of successful IOL (Figure 4). The area under the curve (AUC) of Bishop score (0.718, 95% CI 0.600–0.837, *p* 0.002), CL (0.396, 95% CI 0.269–0.523, *p* 0.130), and UCA (0.658, 95% CI 0.521–0.795, *p* 0.025) were calculated. The cut-off value of UCA was >108.4°. This cut-off value yielded a sensitivity of 69.6%, a specificity of 65.2%, a positive predictive value (PPV) of 81.4%, and a negative predictive value (NPV) of 46.2% for the prediction of successful IOL (Table 2).

The prediction of the performance of the Bishop score showed higher specificity than UCA (cut-off ≥6 in nulliparous/≥5 in multiparous, sensitivity of 60.7%, specificity of 76.9%, PPV of 85.0%, and NPV of 47.6). The odds ratio of UCA was 4.333 (*p* = 0.004, 95% CI 1.612–11.645) and that of the Bishop score was 5.152 (*p* = 0.002, 95% CI 1.788–14.843). In combination analysis with UCA and Bishop score, the higher performance was showed (Bishop ≥ 6 in nulliparous/≥ 5 in multiparous and UCA > 108.4°, as sensitivity of 44.6%, specificity of 96.0%, PPV of 96.2% and NPV of 43.6, OR 10.667 (95% CI 2.446–28.410); Bishop ≥ 6 in nulliparous/≥ 5 in multiparous or UCA > 108.4°, as sensitivity of 85.7%, specificity of 50.0%, PPV of 78.7% and NPV of 61.9, OR 6.000 (95% CI 2.052–17.544)). The mean time of latent phase (time to induction beginning to active phase) in the prior UCA > 108.4° was statistically shorter than the prior UCA < 108.4° group (481min ± 108 vs. 884min ± 141, *p* = 0.009 by Mann–Whitney test). Especially, the ratio of nulliparity and multiparity that estimated the prior UCA > 108.4° was statistically not different (nulliparity, 50.4%, 63/125; multiparity, 62.5%, 50/80, *p* = 0.087 by Fisher’s exact test).

## 4. Discussion

Because IOL is a significant issue in obstetrics, evaluation of pre-delivery cervix status is an important consideration. Although the Bishop score has been accepted as a useful predictive tool, more objective and convenient predictive approaches are needed. As such, research is currently focused on prediction using sonographic findings [28,29]. This study investigated the value of UCA, CL, and the traditionally used Bishop score for predicting IOL. Our findings demonstrate that the ability of UCA in predicting IOL is comparable to the Bishop score.

A shorter CL may suggest more successful labor induction based simply on assuming that the same force is exerted in a shorter distance. However, although CL is short, the efficiency of force may change depending on how the vector applied to the labor force is transmitted. The predictable component of this force vector is the UCA; if this angle is acute, the vector may distribute the force, making it smaller than the original labor force. Conversely, if the angle is obtuse, the vector may not distribute the force, making it similar to the original labor force [7]. This hypothesis warrants further investigation.

In this study, UCA and Bishop scores were statistically different between the successful and non-successful IOL groups but CL was not (Table 1). Although CL is the most widely used sonographic parameter to evaluate the cervix, recent systematic reviews and meta-analyses report both pros and cons of using CL to predict IOL. Hatfield et al., in a meta-analysis, revealed that CL was not an effective predictor of successful IOL. However, because the definition of IOL varies in the literature, CL has been found to successfully predict IOL in terms of ripening [28]. Smith et al. performed a similar meta-analysis and found that CL at or near term was moderately effective in predicting IOL [14]. In both reports, CL was not superior to the Bishop score; therefore, more comprehensive methods integrating both sonography and digital exam may be appropriate.

In this study, the ability of UCA was not superior to the Bishop score (Table 2). The cut-off value for the Bishop score was calculated as 6, and this was commonly found in other literature as a favorable cervix score [29,30]. However, Bueno et al. suggested a different cut-off between nulliparous and multiparous. Therefore, we selected ≥ 6 as the cut-off in nulliparous and ≥ 5 as the cut-off in multiparous women [31]. Unlike the Bishop score, UCA did not contain outlier data in the interquartile range (IQR) (Figure 3). This may be since the Bishop score has limitations of both subjectivity and of being a nominal scale (0–10); therefore, outliers might be found more frequently than in UCA. This reveals another advantage of using UCA as a reproducible and objective parameter rather than the Bishop score.

The mean time of the latent phase was statistically shorter in the wider UCA group. In a previous study, Dagdeviren et al. reported successful IOL based on entering the active phase within 24 h. In our study, wider UCA suggested early active-phase entry within 12 h. rather than 24 h. We thought this might be useful when having a discussion with the mother and the family, who are often concerned about labor and delivery. Also, the ratio of nulliparity and multiparity was not different between the UCA-based IOL success group and the non-success group. The parity did not show a statistical correlation with the UCA-based IOL success group, and, therefore, UCA may be independent of parity. Similar to the results from our study, Ozkaya et al. reported that UCA is a predictor of a satisfactory response to labor induction, especially under 12 h latent phase regardless of parity [23]. Because UCA is based on basic physics, i.e., using the summation of vectors in both directions from the anterior wall and endocervical canal, it is considerably different from the Bishop score which is based on cervical position or effacement [7].

A key limitation of this study was the small sample size due to few cases of labor induction in normal term pregnancies in this tertiary center. The inter- or intra-observer variation when assessing the Bishop score and UCA might also be considered another limitation. In this study, while the same sonographer measured the UCA, the Bishop score was assessed by multiple OBGYN residents. This might also have introduced bias affecting the performances of both UCA and Bishop score prediction. Third, low AUC of UCA made it difficult to evaluate the superiority of UCA compared to the Bishop score. Finally, the lack of mode of delivery or complications is a limitation when evaluating various pregnancy-related outcomes.

Contrary to the present study, Dagdeviren et al. recently reported that UCA was not a useful predictor of IOL in a well-designed, single-center study [7]. However, these conflicting results might be due to the number of multi-parity pregnancies and differences in IOL definition, as various centers use different protocols and definitions of successful IOL [6]. Moreover, the induction methods were different in the study by Dagdeviren et al., who only used prostaglandin compared to the PGE2 and oxytocin used in the current study. Similar to the findings of this study, a recently published research on second-trimester terminations revealed that UCA was significantly wider in patients who successfully terminated pregnancies, while CL was not significantly different [32]. Further, Dziadosz et al. compared the predictive performance of UCA in spontaneous preterm birth and found values > 105° had higher predictive performance than CL of 25 mm when predicting labor in patients <34 weeks of gestation [19]. Our results hint at the usefulness of UCA in full-term IOL. Keepanasseril et al. proposed a novel scoring system to predict successful IOL using parity, CL, and posterior cervical angle [21]. In our study, the Bishop score and UCA showed a significantly higher possibility in predicting IOL than CL. In comparison between UCA and Bishop score, UCA was not more beneficial than the Bishop score. However, a combination of Bishop score with UCA was found to have better sensitivity and PPV when predicting IOL. Further multicentric studies including UCA and other cervical parameters will be required to gain a better understanding.

In conclusion, UCA was not superior to the Bishop score in predicting IOL. However, UCA combined with Bishop score showed higher performances and might help predict successful IOL as well as the feasibility of labor induction.

## Figures and Tables

**Figure 1 jcm-10-02033-f001:**
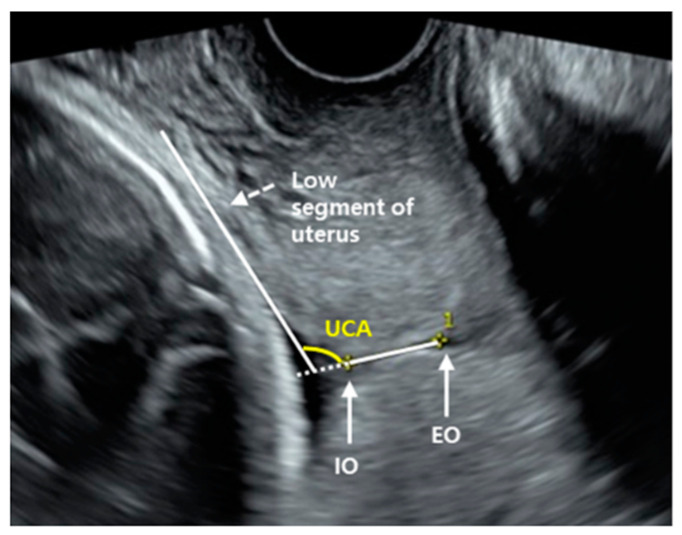
Measuring of uterocervical angle. (IO, internal orifice; EO, external orifice; UCA, uterocervical angle).

**Figure 2 jcm-10-02033-f002:**
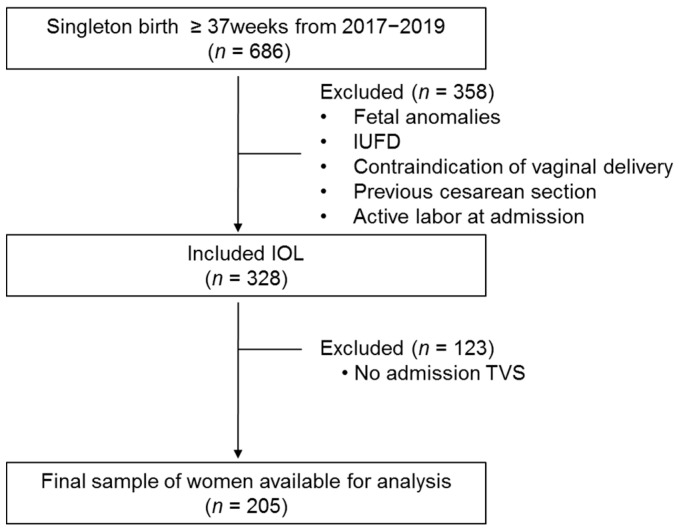
Flowchart of participants inclusion; TVS, transvaginal sonographic.

**Figure 3 jcm-10-02033-f003:**
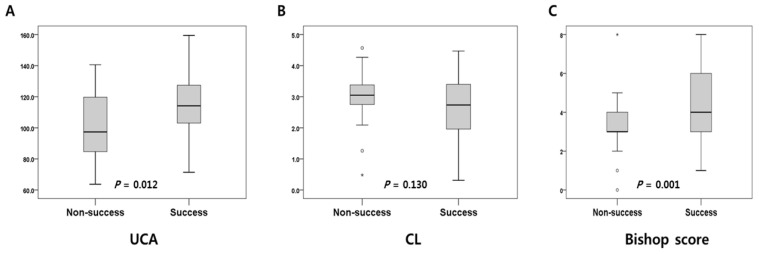
Value distribution of UCA (**A**), CL (**B**), and Bishop score (**C**). The boxes represent the interquartile range (IQR), with the upper and lower edges of the boxes representing the 75th and 25th percentiles, respectively. The central horizontal lines within the boxes represent the median levels for each group. The vertical whiskers above and below the boxes represent the range of outlying data points (up to 1.5 times the IQR). Asterisk(*): potential outlier; zero (°): outlier.

**Figure 4 jcm-10-02033-f004:**
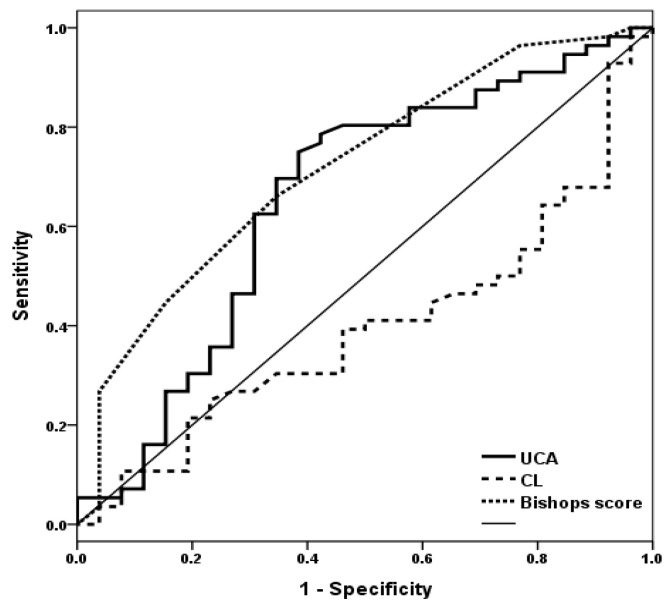
Receiver-operating characteristics (ROC) curve for UCA, Bishop score, and CL to identify the optimal cut-off level for the prediction of the successful induction of delivery. This curve had the following area under the curve (AUC) values: 0.658 for UCA, 95% CI 0.521–0.795, *p* 0.025; 0.718 for Bishop score, 95% CI 0.600–0.837, *p* 0.002; 0.396 for CL, 95% CI 0.269–0.523, *p* 0.130.

**Table 1 jcm-10-02033-t001:** Clinical characteristics of the study population.

Characteristics		Total (*n* = 205)	Non-Success (*n* = 65)	Success (*n* = 140)	*p*
Parity (*n*, %)	Nulliparous	125 (61.0)			
	Multiparous	80 (39.0)			
Maternal age (years)		32 (24–42)	33 (27–41)	32 (24–42)	0.278
BMI (kg/m^2^)		26 (15–32)	26 (15–31)	25 (20–32)	0.643
Gestational age at delivery (weeks)		39 (36–41)	39 (37–41)	39 (36–41)	0.834
Bishop score		4 (0–8)	3 (0–8)	4 (1–8)	0.001
Cervical length (cm)		2.87 (0.31–4.57)	3.05 (0.48–4.57)	2.73 (0.31–4.47)	0.130
UCA (degree)		112.5 (63.7–159.5)	97.3 (63.7–140.6)	114.1 (71.4–159.5)	0.012
Cesarean delivery (*n*, %)		30 (14.6)	20 (30.8)	10 (7.1)	0.001
Indication of induction					
At ≥39 weeks elective induction		118 (52.6)	30 (25.4)	88 (74.6)	
Pregnancy-induced hypertension		18 (2.9)	7 (38.9)	11 (61.1)	
Diabetes		24 (6.8)	11 (45.8)	13 (54.2)	
Fetal macrosomia		9 (3.9)	5 (55.6)	4 (44.4)	
Fetal growth restriction		15 (8.7)	5 (33.3)	10 (66.7)	
Oligohydramnios		12 (5.8)	5 (41.7)	7 (58.3)	
Others		9 (4.3)	2 (22.2)	7 (77.8)	

Values are expressed as median (range) unless otherwise indicated. A *p*-value of <0.05 with a 95% confidence interval was considered significant. Analysis was by Mann–Whitney U test except for cesarean delivery (chi-square test). Abbreviations: BMI, body mass index; UCA, uterocervical angle.

**Table 2 jcm-10-02033-t002:** Performance of uterocervical angle, Bishop score, and combined for predicting successful induction.

	UCA	Bishop Score	Combined (AND)	Combined (OR)
Sensitivity (%)	69.6	60.7	44.6	85.7
Specificity (%)	65.2	76.9	96.0	50.0
PPV (%)	81.4	85.0	96.2	78.7
NPV (%)	46.2	47.6	43.6	61.9
OR (95% CI)	4.333 (1.612–11.645)	5.152 (1.788–14.843)	10.667 (2.446–28.410)	6.000 (2.052–17.544)
*p*-value	0.004	0.002	<0.001	0.001

Abbreviations: UCA, uterocervical angle; OR (95% CI), odds ratio (95% confidence interval).

## Data Availability

The data presented in this study are openly available.

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
