# Peer review of "The Uterocervical Angle Combined with Bishop Score as a Predictor for Successful Induction of Labor in Term Vaginal Delivery"

_jcm, 2021, doi:10.3390/jcm10092033_

Round 1
Reviewer 1 Report
Thank you for letting me review this paper that investigated an alternative approach to the Bishop Score to determine chances for active labour after induction.
My comments as per below;
Introduction
1) The authors state that labour is induced in 30-40% of women. This varies across the world and a wider estimate should be chosen. The same accounts for failure of induction. In the latest RCTs of induction of labour the success rate is higher than 70% and reaches about 80%.
2) The authors state that the Bishop score has high inter-observer variability and references a review. Would the authors explain where the original articles originate from behind this statement?
3) The authors state that ultrasound measurements of cervical length are reproducible. There are papers stating the opposite (such as PMID: 32392356 where it is concluded that "Agreement and reliability of cervical length measurements differed substantially between examiner pairs and examiners". Could the authors elaborate a little more on the reproducibility of TVS and where that originates from?
4) CL is predominately a tool to rule out PTB rather than rule in. The authors argue that UCA was a better predictor for PTB. Does this account for both positive and negative predictive value?
Methods
1) Would be nice to see a flowchart of the proportion of women that were excluded for the various reasons.
2) Was there any training for the sonographers that preformed the measurements? As stated in the previous paper (PMID: 32392356) it was common with intra and inter observer variation even though these were trained ultrasonographers. Were there any evaluation of inter and intra observer variability in the study? Was there a specific instruction on how to measure? Not only interpreting the saved images but how to place the probe and what structures to include in the image.
3) Why was only one monographer chosen to evaluate the measurements? To ensure low inter-observer variation, there has to be two people measuring the same cervix.
4) Who defined onset of labour? Midwife? Doctor? How was it assessed? Definition? How was the 12 hour cut off chosen?
5) Were there a power calculation done to investigate the sample size needed to detect x% difference in precision of predicting successful onset of labour between the methods? I can't see it in the manuscript.
6) Why was this outcome chosen and not mode of delivery and neonatal outcome?
Results
1) Table 1. What were the indications for induction? I generally think Table 1 is not needed on its own it can be combined with table 2 to one table.
2) Figure 2: Bishop score seems to have a higher AUC than UCA and is more widely practiced and in addition, doesn't require an ultrasound machine. Why would then UCA be an alternative? Also, have the authors tried, or has anyone previously tried to combine the measurements into one prediction model in a regression analysis?
3) It is proposed that a mature Bishop score is higher in nulliparous than porous women. Have the authors tried to investigate this in the current material?
4) Line 209: Phage should be changed to phase
5) You state that oxytocin was used for induction in this study. Was that true for all women independent of cervical status? This is not commonly the case in unripe cx where cervical ripening with foley's (cooks) or prostaglandins are usually the methods of choice. Could the authors elaborate on this?
6) The authors conclude that UCA might be an important tool in determining the prediction of successful labour. I would state that the AUC<0.7 would speak against this statement.
Author Response
Very thanks for your opinion and kind comment to our work. We prepare the best answers to your comments. We hope your positive response.
Sincerely,

Reviewer 2 Report
In this retrospective study, the authors assessed the use of uterocervical angle as a predictor of induction of labor success. Based on a cohort of 205 singleton term pregnancies undergoing labor induction, the authors found that induction of labor success was associated with UCA of >108.4 degrees. They conclude that UCA can serve as a sonographic tool to predict labor induction success.
While this study focuses on a rather novel sonographic tool to assess the cervix before labor induction, some major methodological limitations arise, limiting its clinical applicability.
Following are my comments.
Introduction
Line 37- please rephrase the sentence
Lines 41-42- this sentence is general and should be rephrased. Women with unfavorable cervix are undergoing induction as well, and mode of induction, should it be cervical ripening or oxytocin induction, relies on the initial cervical status.
Lines 51-52- is this the authors’ opinion? This statement is not evidence based and not supported by the reference provided (15). Please rephrase.
Two papers evaluated the usefulness of UCA as a predictor of successful induction of labor- pmid 29909269 and 30249147, and should be addressed in the introduction.
Methods
The numbers of participants and excluded cases should be given in the “results” section, not here.
Do the authors perform UCA measurements routinely? For which purpose, as this predictor has yet to be evaluated? Was this intentional as part of the investigation?
Lines 90-91- while one sonographer performed the UCA measurement, how many practitioners assessed the Bishop? This discrepancy should be addressed in the “limitations” section of the study.
Lines 94-95- only oxytocin and PGE2 were used for labor induction? Was cervical ripening part of the induction process? How did the authors define whether cervical ripening was necessary or not?
Line 95- “in appropriately”- please rephrase.
Line 98- how was “active labor” defined? As this is the main outcome measured, a very precise definition should be given here. Why was 12 hours chosen as the time frame for considering failed induction? A large proportion of women will enter active labor more than 12 hours after induction of labor start. Did the authors consider evaluating additional time frames?
Results
Line 111- “discrived” should be corrected
Table 1- a cut-off of 9 hours is provided, discriminating between success and failure of induction. This cut-off is not described by the authors previously, nor previously described as an agreed definition of induction failure.
Table 2- if only 30.8% of women underwent cesarean delivery in the failure group, how did 69.2% of cases deliver? And why was the definition of “failure” used in these cases?
Line 167- the superiority of either method of assessment used can be defined by direct comparison. Was such a comparison performed? Otherwise, please rephrase.
Table 3- How was the 108.4 angle identified? This should be described in the “Methods” section. Most commonly, a Bishop score of ≥6 is considered for the prediction of successful induction. Why did the authors use ≥ 5 vs. ≤ 4?
Discussion
Lines 184-185- reference is needed
Lines 203-204- “Unlike the Bishop score, UCA did not contain outlying data in the interquartile range (IQR)”- this sentence merits further explanation. Is the data presented in
Figure 1 or 2? If this sentence refers to Figure 1, the number of outliers is relatively small.
Lines 209 and 211- “phage”- please correct
Lines 215-216- please rephrase
Line 223- incorrect reference.
Lines 221-237- there are only two sentences referring to the study’s limitation. There are several other limitations that should be addressed by the authors. Further, the “Limitations” section deserves a separate paragraph, not mixed with the discussion on previous literature.
Furthermore, the authors did not refer to the study by Eser and Ozkaya (pmid 30249147), that reported a correlation between UCA and labor induction success.
There is no referral to Bishop parameters that are not assessed by the UCA such as dilatation, consistency or fetal head station. These should be addressed in the “Discussion”.
Author Response

(The authors gave the same response as above.)

Round 2
Reviewer 1 Report
Thank you for the revised manuscript. My additional comments as per below.
1. Introduction
1.3The authors state that ultrasound measurements of cervical length are reproducible. There are papers stating the opposite (such asPMID:32392356where it is concluded that "Agreement and reliability of cervical length measurements differed substantially between examiner pairs and examiners". Could the authors elaborate a little more on the reproducibility of TVS and where that originates from?
We agree to your opinion and we revised the sentence. As Bishop score, TVS also has the matter of reproducibility. We will revised this paragraph with more references to introduce of reproducibility of TVS and UCA. Also, we add the reference which you suggested on discussion section. Very thanks for your suggestion.
Thank you. I still wonder though about the reproducibility of Bishop score since you couldn't provide any papers of this. If there aren't any studies of reproducibility of Bishop score, this should then be stated and thus it would be difficult to compare inter- and intra observer variability between bishop score and TVS?
You have kept your expression about Bishop score and intra observer variability as a rationale for ultrasound in the abstract but I take it as this can not be supported since you can't provide any studies of variability of Bishop score?
You have chosen to keep the high induction rate in the introduction with a few references. Since this is very different across the world, you should revise this sentence.
Regarding your statements of "significant discomfort" this would need some clarification. Is it really true that women experience TVS as an investigation with less discomfort than vaginal exam with BS and what is the significance? Now when we discussed that there is no diagnostic advantage in your method over BS, the only possible advantages left are less discomfort and perhaps reduced inter observer variability. But the latter is not confirmed neither in studies nor in the outcome (that lesser intra and inter observer variability would results in a better prediction). You would need to revise the text to make it more balanced in the light of above.
2. Methods
2.2Was there any training for the sonographers that preformed the measurements? As stated in the previous paper (PMID:32392356)it was common with intra and inter observer variation even though these were trained ultrasonographers. Werethere any evaluation of inter and intra observer variability in the study? Was there a specific instruction on how to measure? Not only interpreting the saved images but how to place the probe and what structures to include in the image.Why was only one monographer chosen to evaluate the measurements? To ensure low inter-observer variation, there has to be two people measuring the same cervix.
We surely agree to your opinion. In our hospital, one OBGYN sonographer was enrolled. She is the member of American registry for diagnostic medical sonography (ARDMS) from 2011and has career for 15yrs (2006-2021). All measured CL is followed by Guideline and measuring UCA is confirmed by corresponding author. Although in the previous references (PMID: 32392356) reported measuring CL is required training to adjust inter-observer variation, Ninlapa al reported that UCA measurements had a higher intra-and inter observer reproducibilitythan CL. The measuring protocol is mentioned in material section. The first angle caliper was placed from the external to the internal orifice of the uterus, and the second caliper was then extended along the length of the lower uterine segment.We added new figure 1 of our measurement and described in discussion section.
Thank you for your explanation. In the text it reads
"For UCA measurement, one sonographer was used to evaluate all cases to insure standardization of measurement and to prevent inter-observer bias."
I don't understand this. Would this be a strenght? This would not be feasible in a clinical setting and is rather a limitation and could be something that reduces its reproducibility. It should then be discussed as a limitation in the discussion.
2.3 Who definedonset of labour? Midwife? Doctor? How was it assessed? Definition? How was the 12 hour cut off chosen?
Thanks for your comment. All labor procedure was performed by OBGYN resident doctor and notified to corresponding author. As mentioned in manuscript, success or failed IOL is vary in several studies. In this study, we refer to several studies which defined success of IOL as entering active phase under 12hr (reference: J KoreanMed Sci 2007; 22: 722–727; J ObstetGynaecol Res 2009; 35: 301–306.; J MaternFetal Neonatal Med. 2020 Apr;33(8):1295-1301.)We added these references in manuscript.
Thank you. Could you then please explain the definition of active labour in your population as this can differ between countries. Also, BS of 4 is usually regarded as a low BS where further treatment with prostaglandins or foley's is recommended before oxytocin. Why was oxytocin administered at such an early BS? This should be discussed in the limitations section as the induction procedure then might not be generalizable.
2.5Why was this outcome chosen and not mode of delivery and neonatal outcome?
Thanks for your suggestion. As you commented we considered relation between UCA and mode of delivery. However, the total enrolled population did not sufficient to analyze. Weagree to your opinion that neonatal outcome is not necessary in this result. If you think it is better to remove,please let us know and we revised the table. Thanks for your kindness comment.
I would prefer if you could add this to limitations and acknowledge this as a proxy outcome and not as the important clinical outcome when evaluating the cx for induction.
3. Results
3.1 Table 1. What were the indications for induction? I generally think Table 1 is not needed on its own it can be combined with table 2 to one table.
Indication of induction is all singleton without exclusion criteria.Inclusion criteria for the study were singleton pregnancy, intact membrane, cephalic presentation, absence of active labor at admission. Exclusion criteria is the ofwomen with congenital fetal anomalies, intrauterine fetal death, contraindications to vaginal delivery, previous cervical surgery, refusal to participate, active labor at admission and fetal distress before active labor.As you suggested, we added figure 1 with exclusion criteria. Also, we revised table 1 & 2 to combine. Thanks for your comment.
Thank you. I would prefer if you could characterize your own population regarding indication for induction as well since the success rate of induction might differ on indication (for example, was post-term pregnancy the most common indication? Or women's wish?)
3.2Figure 2: Bishop score seems to have a higher AUC than UCA and is more widely practiced and in addition, doesn't require an ultrasound machine. Why would then UCA be an alternative? Also, have the authors tried, or has anyone previously tried to combine the measurements into one prediction model in a regression analysis?
We agree to your opinion. As you commented, AUCof Bishop score is superior then UCA. Frankly speaking, we expected the better AUC of UCA than Bishop score but it was not. However, with in terms of subjectivity, we thought UCA is worth to consider in IOL. Also, in outpatient setting, pelvic exam is getting discomfort to mother so UCA is the candidate to predict successful IOL and consider admission. Also, in the performance test, UCA is better sensitivity than Bishop so we thought it may be useful. As you suggested we combined the higher UCA and higher Bishop and this showed highest specificity and positive predictive value(UCA ≥108.4+ Bishop ≥6: sensitivity of 30.4%, specificity of 100%, PPV of 100%, NPV40%, Odds 1.587 [95% CI 1.208-1.707]). However, because of not enough sample size we thought it will be necessary further study with largest enrollment.
Perhaps this could be added then as future perspectives, that even if this angle was not superior (rather inferior) to BS, it might be additive in a prediction model that would require a larger sample size? As earlier stated, your conclusion is now not balanced towards your findings since this method is not superior to BS and also requires equipment and trained staff. For example, this is a rationale against TVS in predicting PTB, in favor of vaginal biomarkers.
3.3It is proposed that a mature Bishop score is higher in nulliparous than porous women. Havethe authors tried to investigate this in the current material?
Thanks for your comment. We calculate the Bishop score and score of nulliparous was lower(nulliparity; 3 (0-6), multiparity; 4(1-8); p 0.002). If you suggested that it is better to add in manuscript, let us know and we will correct.
I'm sorry if this was misunderstood. What I meant was that the classification of an inmate BS is often different in parous vs nulliparous where a BS of 6 is considered mature in multiparous and this is usually 7 in nulliparous for example.
3.5You state that oxytocin was used for induction in this study. Was that true for all women independent of cervical status? This is not commonly the case in unripe cx where cervical ripening with foley's (cooks) or prostaglandins are usually the methods of choice. Could the authors elaborate on this?
3.6The authors conclude that UCA might be an important tool in determining the prediction of successful labour. I would state that the AUC<0.7 would speak against this statement.
As you commented in above question, we also expected higher AUC of UCA but it was not. So we didnot thought UCA is superior then Bishop score. However, with higher sensitivity we thought UCA is considered in clinical setting where pelvic exam is not available. We added yourcomment and the answer in limitation section, Thanks for your comment.
See earlier comments. The final statement
"In conclusion, UCA may be a useful sonographic tool for predicting whether successful IOL is feasible when labor induction is performed. "
needs to be revised and also in the abstract. This is not, according to your results, true. Rather, this method has no advantage over BS in your study and the only potential addition could be in a prediction model
Author Response
First of all, we specially appreciate to you for thoughtful suggestions and insights. The manuscript has benefited from these insightful suggestions.
The manuscript has been rechecked with English proofreading, and the necessary changes have been made in accordance with your suggestions.
Thank you for your consideration. I look forward to hearing from you.
Sincerely,

Reviewer 2 Report
Overall, the authors responded only partially to the comments raised. Furthermore, thorough English editing is still needed.
Lines 36-37- this sentence is not related to the next one, should be rephrased
Lines 39-40- has to be rephrased
Lines 55-56- this is a conclusion that is not evidenced-based. If citing a reference, the authors should clarify it
What does this study add in relation to previous studies?
Lines 116-117- this is merely a description of the sample size, and not the calculation performed.
Lines 254-255- the sentence is not clear
Lines 262-278- this paragraph mixes limitations and interpretation of the current study with a previous study.
Author Response

(The authors gave the same response as above.)
